

# Diel fluctuations of viscosity-driven riparian inflow affect streamflow DOC concentration

Michael P. Schwab[1,2], Julian Klaus[1], Laurent Pfister[1], Markus Weiler[2]

[1]Catchment and Eco-Hydrology Research Group, Luxembourg Institute of Science and Technology, Belvaux, 4422, Luxemburg
[2]Hydrology, Faculty of Environment and Natural Resources, University of Freiburg, Freiburg, 79098, Germany

*Correspondence to*: Michael P. Schwab (michaelschwab.fr@gmx.de)

**Abstract.** Diel fluctuations of streamwater DOC concentrations are generally explained by a complex interplay of different instream processes. We measured the light absorption spectrum of water and DOC concentrations *in-situ* and with high-frequency by means of a UV-Vis spectrometer during 18 months at the outlet of a forested headwater catchment in Luxembourg (0.45 km$^2$). We generally observed diel DOC fluctuations with a maximum in the afternoon during days that were not affected by rainfall-runoff events. We identified an increased inflow of terrestrial DOC to the stream in the afternoon, causing the DOC maxima in the stream. The terrestrial origin of the DOC was derived from the SUVA-254 (specific UV absorbance at 254 nm) index, which is a good indicator for the aromaticity of DOC. In the studied catchment, the only possible process that can explain the diel DOC input variations towards the stream is the so-called viscosity effect. The water temperature in the upper parts of the riparian zone is increasing during the day, leading to a lower viscosity and therefore a higher hydraulic conductivity. Consequently, more water from areas that are rich in terrestrial DOC passes through the riparian zone and contributes to streamflow in the afternoon. We believe that not only diel instream processes, but also viscosity driven diel fluctuations of terrestrial DOC input should be considered for explaining diel DOC patterns in streams.





## 1 Introduction

Dissolved organic matter (DOM) is a major constituent of the carbon cycle and aquatic biogeochemistry, eventually linking terrestrial and aquatic ecosystems (Battin et al., 2008; Lee et al., 2016; Saraceno et al., 2009). The largest component of DOM in forested stream ecosystems is dissolved organic carbon (DOC) (McLaughlin and Kaplan, 2013). DOC has a

multifaceted chemical character that is mainly determined by its origin and its biogeochemical transformation (Hanley et al., 2013; Ruhala and Zarnetske, 2017). DOC in streams is mainly derived from external terrestrial sources (allochthonous) like plants and soils or from instream microbial sources (autochthonous). With increasing stream orders, autochthonous sources become more important (Dawson et al., 2001; Nimick et al., 2011). While DOC from allochthonous sources is characterized by fulvic and humic acids with high molecular weight and aromaticity, DOC from autochthonous sources has a lower

molecular weight and is less aromatic (Hood et al., 2006; Saraceno et al., 2009; Spencer et al., 2012).

Different techniques have been used to gain information on the composition and the concentration of DOC. Two frequently used optical methods to characterize bulk DOC are UV-Vis spectroscopy and fluorescence spectroscopy (Minor et al., 2014). For identifying the aromaticity of DOC in aqueous systems, the specific UV absorbance at 254 nm (SUVA-254) is a commonly used index. SUVA-254 is calculated as the UV absorbance of water at the wavelength of 254 nm (A254) that is

normalized for DOC concentration (Weishaar et al., 2003). A higher SUVA-254 value indicates a higher aromatic DOC content and is therefore a valuable index for distinguishing between allochthonous and autochthonous origins of DOC.

Several studies used SUVA-254 to identify DOC from different origins in combination with changing contributions from different water sources and flowpaths. Hood et al. (2006) observed an increase of SUVA-254 during a 6-day storm event in three catchments of the H.J Andrews Experimental Forest, Oregon (USA) and suggested SUVA-254 as a useful tracer for

identifying different flowpaths through mineral soils. Also at HJ Andrews, Lee et al. (2016) observed lower SUVA-254 values during the dry season low flows and suggest, supported by fluorescence indices, that in those conditions the stream water originates from more microbially-processed sources. Fasching et al. (2016) described similar observations in an Austrian, alpine second-order stream. They related the increase in SUVA-254 values during high flows mainly to a rise in terrestrial DOC contributions. Likewise, they correlated the decrease in SUVA-254 values during baseflow conditions to

larger contributions from autochthonous DOC sources. As an alternative, Catalan et al. (2013) identified seasonality as the main factor controlling SUVA-254 patterns in an ephemeral Mediterranean catchment, because vegetation is accumulated during the dry period. In comparison to mechanistic studies focusing at seasonal and event timescales, investigations combining diel DOC fluctuations with SUVA-254 calculations are rather scarce. While Fasching et al. (2016) did not find clear diurnal SUVA-254 patterns in their stream, they were able to document diel DOC fluctuations with recurrent maxima

around 19:30 h. They linked this pattern to a decrease in Photosynthetically Active Radiation (PAR).

Diel DOC fluctuations in streams are generally explained by a complex interplay of different instream processes. They cannot be observed in every stream, but when they occur, DOC concentrations are often increasing during daytime and decreasing at night (Nimick et al., 2011 and reference therein). Throughout daytime, autotrophic organisms like algae



excrete labile DOC during their photosynthesis, which depends on stream temperature and the amount of sunlight. On the contrary, more instream DOC is consumed at night by heterotrophic organisms (Chittoor Viswanathan et al., 2015; Fasching et al., 2016; Nimick et al., 2011; Parker et al., 2010; Spencer et al., 2007). This interplay of autotrophic and heterotrophic organisms is generally used to explain diel DOC fluctuations in streams. Other studies observed diel DOC fluctuations with

DOC maxima in the early morning due to the absence of photic removal processes of DOC during the night (Worrall et al., 2015; Worrall and Moody, 2014). Tunaley et al. (2017) observed DOC maxima in the early morning for a peatland catchment, whereas a proximate catchment had its DOC maxima in the afternoon. Spencer et al. (2007) reported two DOC maxima per day in the San Joaquin River (California, USA).

In our study, we observed diel DOC concentration fluctuations at the outlet of a 0.45 km$^2$ forested headwater catchment.

Throughout the year, the maximum diel DOC concentrations occurred in the afternoon during baseflow conditions. Based on our literature review of mechanistic explanations of DOC fluctuations our first hypothesis states that diel fluctuations in DOC concentrations are controlled by instream microbial processes. Our second hypothesis stipulates that diel fluctuations in DOC concentrations can be explained by an increased input of terrestrial DOC to the creek during daytime. This second hypothesis is a follow-up on previous work by Schwab et al. (2016) carried out the Weierbach catchment. They linked diel

fluctuations in discharge to increased inflow from the riparian zone in the afternoon due to variations in viscosity (viscosity effect). Before the growing season, Schwab et al. (2016) observed diel discharge fluctuations with maxima in the afternoon that can be explained by riparian water temperature fluctuations and therefore viscosity fluctuations. Warmer riparian water temperature in the afternoon led to a lower viscosity of water, resulting in a higher hydraulic conductivity and therefore an increasing inflow of water to the creek when passing through the riparian zone. During the growing season, discharge

minima were observed in the afternoon due to the stronger influence of evapotranspiration. Nevertheless, Schwab et al. (2016) concluded that the viscosity effect was still present during the growing season, but not visible anymore in the diel discharge fluctuations as a result of the increased importance of evapotranspiration. We intend to leverage these findings through our second hypothesis, stating that the viscosity effect could possibly increase the input of terrestrial DOC in the afternoon all year long.

We used SUVA-254 for testing both hypotheses. A decrease in SUVA-254 values during the afternoon would lead to the rejection of the second hypothesis, stating that an augmented input of terrestrial DOC can explain the DOC concentration maxima in the creek. Increased SUVA-254 values would lead to the rejection of the first hypothesis, where instream processes are supposed to control fluctuations in DOC concentrations.

## 2 Methods

We measured the DOC concentration and the light absorption spectrum with a UV-Vis spectrometer in the Weierbach creek in Luxembourg from December 2013 to May 2015. The Weierbach is a headwater catchment with a size of 0.45 km$^2$ and elevations ranging from 450 to 512 m a.s.l. Beech (Fagus sylvatica) and in a smaller part spruce (Picea abies) are the





dominant tree species in this forested catchment. The soils are shallow Cambisols with a depth of generally less than one meter and the bedrock geology consists of Devonian metamorphic slate and overlying Pleistocene Periglacial Slope Deposits (Moragues-Quiroga et al., 2017). In the vicinity of the creek, the hillslopes are gentle on the right bank side and steep on the left bank side, while further uphill slopes tend to plateau. Along most parts of the creek a riparian zone extends up to 3

meters away from the channel and connects the hillslopes to the creek. Water passing through the riparian zone contributes significantly to discharge, both during wet and dry conditions.

At the outlet of the Weierbach catchment, we measured water levels with a pressure transducer (ISCO 4120 Submerge Probe) at 5 minute intervals. Water levels were converted into discharge via a rating curve. We corrected the temperature sensitivity of the probe according to the stream water temperature (Schwab et al., 2016). Precipitation was measured with a

tipping bucket rain gauge at the meteorological station of Roodt, 3.5 km outside the Weierbach catchment. Precipitation had no distinct seasonality and the long term annual average was approximately 950 mm. During the observation period, no substantial snowfall was observed. The annual rainfall runoff ratio was around 50% with higher discharge volumes in winter than in summer (Glaser et al., 2016; Martínez-Carreras et al., 2015; Pfister et al., 2017; Schwab et al., 2016).

In one part of the riparian zone with high subsurface flow to the creek, we measured the riparian groundwater temperature

every 30 min at 10 cm depth. We calculated the viscosity of the riparian water according to the Vogel equation (Schwab et al., 2016; Vogel, 1921). An increase of water temperature by 5 °C leads to a decrease in viscosity by 12 % to 15 % and therefore to an increase in hydraulic conductivity in the same range (Tipler and Mosca, 2008).

The DOC concentrations and the light absorption spectrum were measured *in-situ* in the Weierbach creek at an interval of 15 minutes with the UV-Vis spectrometer spectro::lyser (s::can Messtechnik GmbH). The spectrometer measured the light

absorption spectrum of the stream water between 220 and 720 nm in 2.5 nm resolution with a xenon flash lamp, 256 photo diodes and a two beam instrument. The optical path length was 35 mm. The spectrometer probe was fixed to a metal plate that was placed on the streambed of the Weierbach creek. The orientation of the probe was horizontal and in stream direction with the measuring window facing towards the riverbed to avoid direct solar radiation. Every three hours, the measuring window of the spectrometer probe was cleaned automatically with pressurized air that was produced by an air compressor.

We cleaned the spectrometer manually every two weeks.

We adapted the global calibration of the spectrometer that was provided by the manufacturer of the instrument to the local conditions by applying a local calibration. For this, we manually sampled the stream water weekly to biweekly and took automatic samples during several rainfall events. We analyzed the grab samples in the laboratory for DOC with a combustion analyzer (Apollo 9000 - Teledyne Tekmar) and compared the results with the *in-situ* DOC concentration

measurements of the spectrometer at the collection time of the grab samples. The linear regression for the local calibration between the lab values and the spectrometer values resulted in a good fit with an $R^2$ of 0.96.

A long time series of end-member chemistry data is available for the Weierbach catchment (Martínez-Carreras et al., 2015). DOC concentration values of biweekly sampled end-members are available since 2009, while biweekly UV-absorbance values at 254 nm (A254) are available since 2012. The sampled end-members included throughfall, soil water, riparian water



and shallow groundwater. Throughfall was collected as bulk samples over two weeks at three different locations. Soil water was sampled by applying a vacuum to suction cups that were installed at six different locations in the soil at depths of 10 cm to 100 cm. At one location in the riparian zone, riparian water was collected with the same method. The biweekly grab samples of shallow groundwater were pumped from three wells in the catchment. The wells were screened for the lowest 50

cm to one meter and had a depth of two to three meters.

SUVA-254 is a commonly applied index for characterizing the aromaticity and the terrestrial origin of DOC. SUVA-254 (l $mg^{-1}$ $m^{-1}$) is calculated as the UV absorbance at 254 nm (A254 in $m^{-1}$) divided by the DOC concentration (mg $l^{-1}$) (Weishaar et al., 2003). For the SUVA-254 data of the end-members, A254 and the DOC concentrations of the biweekly grab samples were measured in the laboratory. To calculate the high-frequency SUVA-254 values of the stream water, we used the *in-situ*

spectrometer measurements of DOC and the light absorbance measurements. Due to the 2.5 nm intervals of the spectrometer, the absorbance data at 254 nm (A254) was not available. Therefore we calculated A254 as the weighted mean between the absorbance at 252.5 nm and the absorbance at 255 nm. We eliminated potential outliers in the SUVA-254 time series by applying a 3 hours moving median to the entire time series.

For analyzing the diel fluctuations of DOC concentrations, SUVA-254, viscosity and discharge, we selected the days with

diel fluctuations during the observation period from December 2013 to May 2015. Days that were influenced by rainfall-runoff events were not included in the analysis. According to this criteria, many short and several longer periods were removed. Especially the two winter seasons and August 2014 were particularly rainy periods. From the remaining days, additional days were removed from further analysis if at least one of the four variables showed unreliable or no values, especially due to problems with the used sensors. A longer period had to be removed in October 2014 because of that same

reason.

We first analyzed the diel fluctuation patterns of DOC, SUVA-254, viscosity and discharge by comparing their daily minima, maxima and amplitude. For each day with diel fluctuations, we calculated the time of the day, when the minima and maxima occurred. The daily amplitude resulted from the difference between the values of the daily maximum and minimum. For further analysis, we calculated the anomaly of each of the four variables, DOC, SUVA-254, viscosity and discharge,

around their daily moving average from the original time series with the 15 minute time intervals. The daily moving average was calculated from the original time series with a window size of 24 hours and did not show diel fluctuations anymore.

We studied the anomalies of the four variables by comparing them with the corresponding values at the same time of another variable using scatterplots. With four different variables (DOC, SUVA-254, viscosity, discharge), this resulted in six different combinations. For each combination, linear regressions were calculated separately for each month, for the dormant

and growing season and for the entire observation period. Due to the absence of days with diel fluctuations, we could not compute a linear regression for January. We defined the growing season as the period between the 15th of May and the end of September and the dormant season from the beginning of October until the 15th of April. To clearly distinguish between the two seasons, we introduced a transition period. As a transition period, we considered the time between mid-April and




mid-May, when not all plants are yet fully active and developed. A transition period was not defined in fall, due to the lack of days with diel fluctuations around the end of September and the beginning of October.

## 3 Results

In our long-term high-frequency time series, we observed many days and periods with diel fluctuations in viscosity, SUVA-254, DOC and discharge. In the afternoons of rainless periods during the dormant and the growing season, we observed the diel minima of viscosity and the diel maxima of SUVA-254 and DOC (Figure 1). During the dormant season, we observed diel discharge minima in the morning, whereas we observed diel discharge minima in the afternoon during the growing season. The diel amplitudes of viscosity, SUVA-254 and DOC are changing in similar ways from one day to the other (Figure 1 e-g).

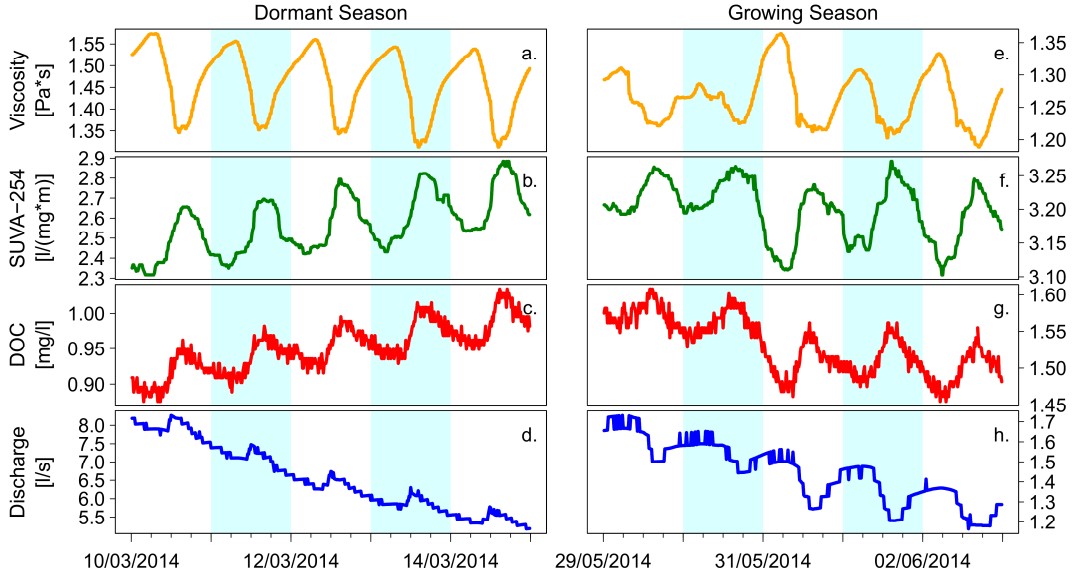

**Figure 1: Diel fluctuations of viscosity of riparian water, SUVA-254, DOC and discharge during a representative rainless period in the dormant season and the growing season.**

Over the whole time series of 18 months, the minima in viscosity and the maxima in SUVA-254 and DOC occurred in the afternoon between 14:00 h and 18:00 h for both, the growing and the dormant season (Figure 2 a-c). For discharge, the time of the minima switched from early morning in the dormant season to the afternoon in the growing season both in 2014 and 2015 (Figure 2d). In winter, we only observed a few rainless days outside rainfall-runoff events with diel fluctuations (Figure 2). During that time (December 2013 and November 2014) no clear diel discharge pattern is visible (Figure 2d) and the diel





amplitudes of all four variables are relatively small. The diel amplitudes of DOC and viscosity stayed relatively constant over the 18 months with lowest amplitudes of DOC in winter and spring and slightly higher viscosity amplitudes during the growing season than during the dormant season (Figure 2 a,c). The amplitudes of SUVA-254 changed more markedly over the 18 months. SUVA-254 had its highest amplitudes in spring and very low amplitudes in summer (Figure 2b).

Figure 2 shows a seasonal pattern for the daily mean values of all four variables. The viscosity of the riparian water is lower during the growing season than during the dormant season (Figure 2a), while the mean daily SUVA-254 values and the mean daily DOC concentrations are higher during the growing season than during the dormant season (Figure 2 b,c). The discharge in the Weierbach creek was lower in summer and higher in winter and early spring (Figure 2d).

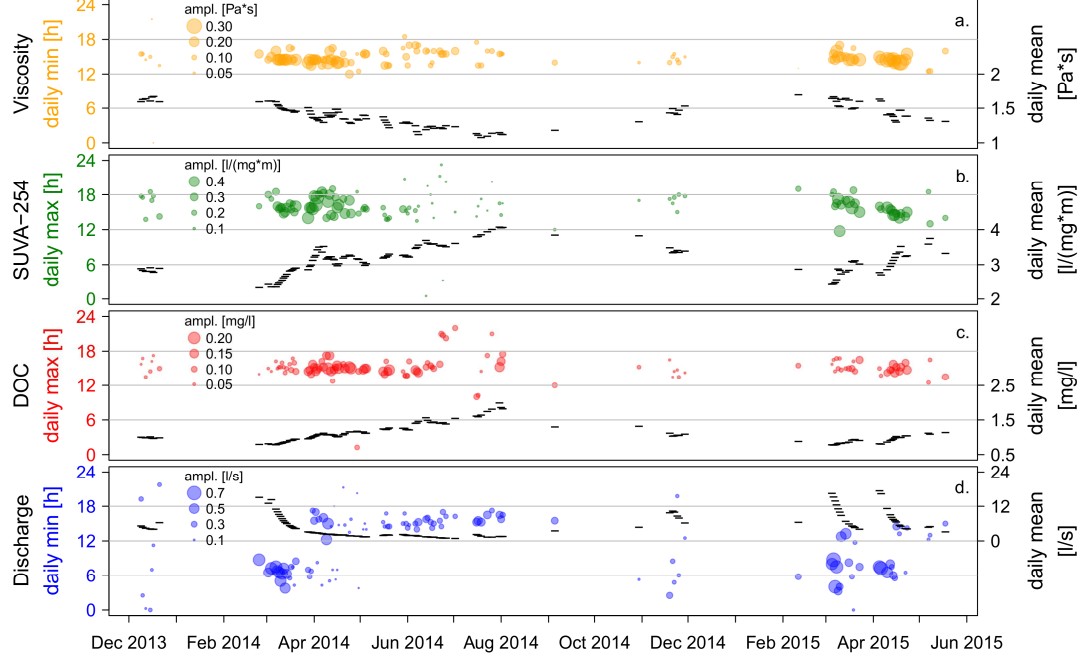

Figure 2: The time of day of the daily minima/maxima of riparian water viscosity, SUVA-254, DOC and discharge over 18 months.
Only rainless days with diel fluctuations and without the influence of rainfall-runoff events are represented. The points are scaled by the daily amplitude between the daily minimum and maximum. Black dashes (-) are the daily mean values of the respective variables.

After identifying the strong similarity in the timing of the diel extreme of viscosity, SUVA-254 and DOC, we analyzed the
relationship between the 15 minute anomalies of viscosity, SUVA-254, DOC and discharge. Figure 3 and Figure 4 show a strong linear relationship between SUVA-254 and viscosity, SUVA-254 and DOC as well as between DOC and viscosity for the dormant season, the growing season and the entire time series of 18 months with $R^2$ larger than 0.6. The slope of the



linear regression between the viscosity anomalies and the SUVA-254 anomalies is negative, meaning that the viscosity of the riparian water was decreasing during the day, while SUVA-254 values were increasing (Figure 3a). During the growing season, the slope was less negative than during the dormant season (Figure 3a and Figure 4a). The values of the slopes show an annual pattern, with the least negative slopes occurring in June and July (Figure 4a). The slope of regression between the

5    DOC anomalies and the SUVA-254 anomalies is positive (Figure 3b). An increase of SUVA-254 during the day leads to an increase in DOC concentrations. This relationship is less strong during the growing season, with the smallest slopes occurring in June, July and August (Figure 3b and Figure 4b). The slope of the regression between viscosity and DOC is negative, meaning that a decrease in viscosity during the day leads to an increase in DOC (Figure 3d). These negative slopes are relatively constant over the year and between the seasons (Figure 3d and Figure 4d).

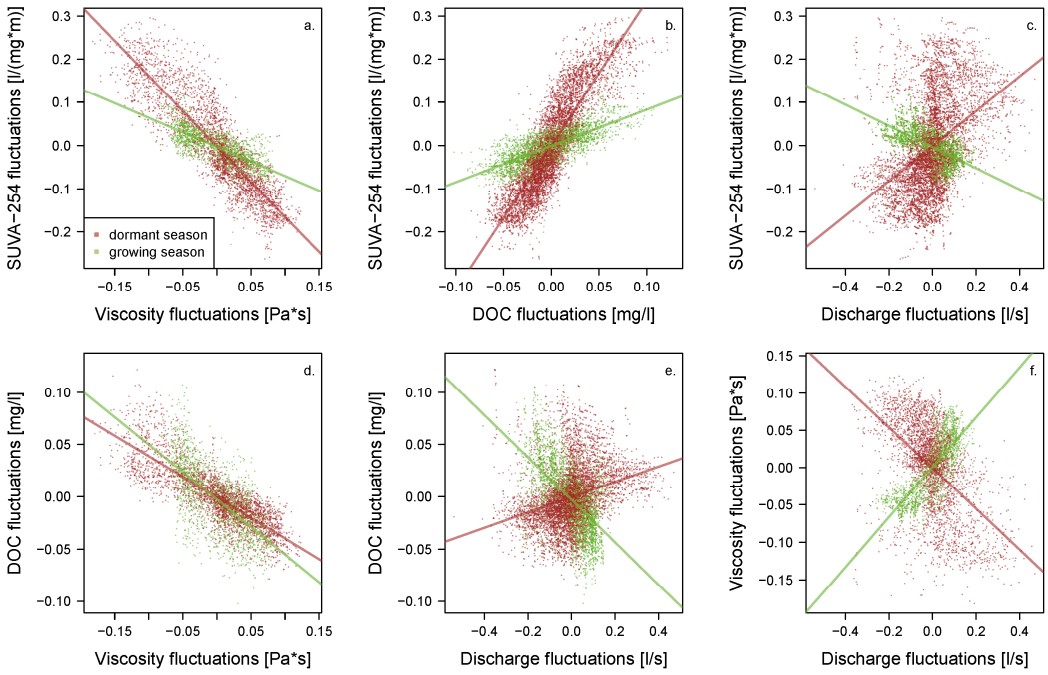

**Figure 3: Scatterplots and linear regression between the 15 minute anomalies of the four variables for the growing and dormant period. Only rainless days with diel fluctuations and without the influence of rainfall-runoff events are shown (corresponding to the days in Figure 2).**

For the combinations that included discharge, we generally observed weaker and more heterogeneous relationships (Figure 3

15    c,e,f and Figure 4 c,e,f). The linear regressions between discharge and SUVA-254, discharge and DOC, as well as between discharge and viscosity resulted in contrary signs of their slopes between the dormant season and the growing season.





Moreover, the $R^2$ of the linear regressions where discharge was involved, were generally smaller than for the linear regressions in absence of discharge (Figure 4).

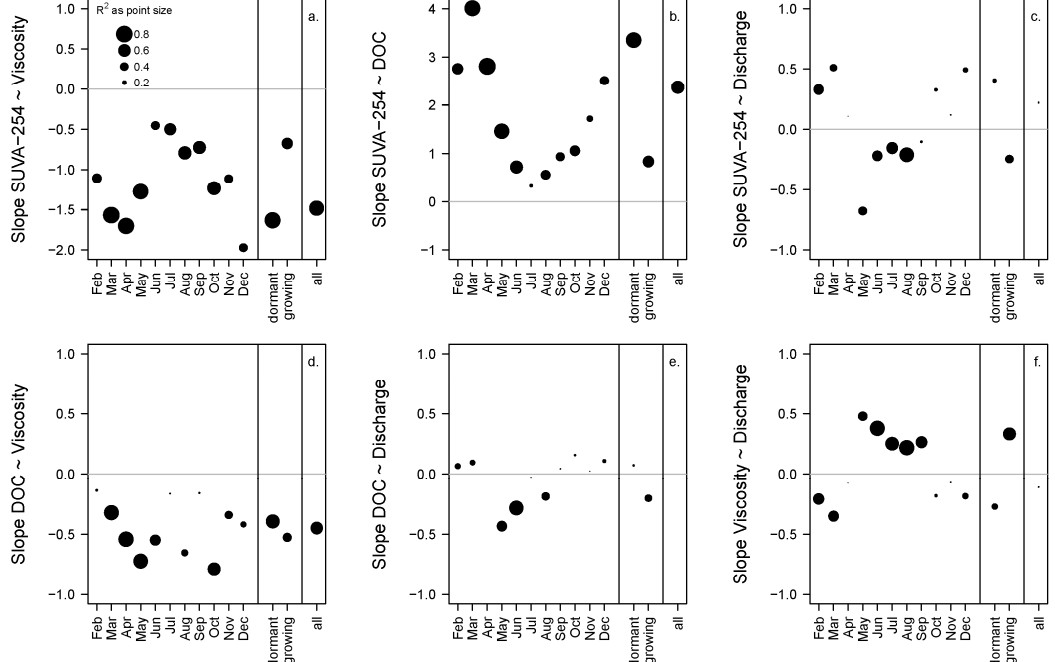

**Figure 4: Slope and explained variance ($R^2$) of the linear regression between the 15 minute anomalies of the four variables. Slope**
**and $R^2$ are separately calculated for each month, for the dormant and the growing season and for all values. All the p-values are generally highly significant. Only rainless days with diel fluctuations and without the influence of rainfall-runoff events are shown (corresponding to the days in Figure 2).**

In addition to the high-frequency instream observations and temperature measurements of the riparian zone, we sampled end-members in the catchment and analyzed them in the laboratory for SUVA-254 and the DOC concentrations. We
observed the highest DOC concentrations in throughfall and soil water, lower concentrations in riparian water and lowest DOC concentrations in the groundwater (Figure 5a). We found a decrease of DOC concentrations in soil with depth. The highest DOC concentrations were observed in the upper part of the soil profile (Figure 5b). The SUVA-254 values in soil water behave similarly to the DOC concentrations, having the highest values in the upper part of the soil profile (Figure 5c). Soil water, throughfall and riparian water have similar SUVA-254 values, while groundwater has the smallest SUVA-254
values (Figure 5d).





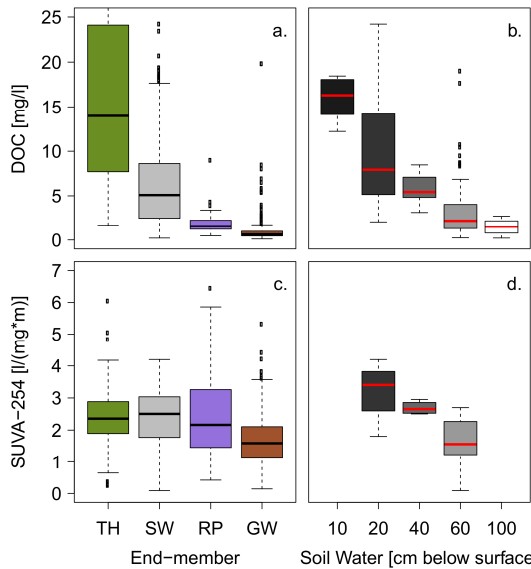

**Figure 5: DOC concentrations and SUVA-254 values of the biweekly sampled end-members and the detailed information for soil water at different depths. TH = throughfall, SW = soil water, RP = riparian water, GW = groundwater.**

## 4 Discussion

Based on our measurements in the Weierbach catchment, we are convinced that SUVA-254 is a suitable proxy for identifying terrestrial DOC in diel DOC fluctuations. Several studies already demonstrated that SUVA-254 is a valid index to characterize the origin of DOC (Catalán et al., 2013; Fasching et al., 2016; Lee et al., 2016; Weishaar et al., 2003). We found strong evidence in the Weierbach catchment for rising SUVA-254 values serving as a valid index of higher terrestrial DOC input to the creek. Immediately after rain events, discharge, DOC concentrations and SUVA-254 rapidly increased.

This increase in discharge is related to surface or near-surface runoff processes (Glaser et al., 2016; Klaus et al., 2015). Therefore it is likely that the increase in DOC concentrations was induced by terrestrial DOC input that eventually led to a rise in measured SUVA-254 values.

We tested our two hypotheses on processes controlling diel DOC fluctuations. For the days with diel fluctuations we generally observed both DOC and SUVA-254 maxima in the afternoon. Thus we could reject our first hypothesis that the

DOC maxima in the afternoon are controlled by microbial autochthonous instream processes. Moreover, the increased SUVA-254 values in the afternoon are a strong support for our second hypothesis that the DOC maxima in the afternoon are triggered by an increase in terrestrial DOC input in the afternoon. Another support for the second hypothesis is that the high-frequency anomalies of DOC and SUVA-254 behave in a similar way as suggested by the good fit of the regression between



those two variables. Additionally, the SUVA-254 values and DOC concentrations of the end-members are a strong indicator of stream water origin in the afternoon (when SUVA-254 and DOC are on the rise). For both DOC and SUVA-254, soil water and riparian water had higher values than groundwater and the values in the topsoil were higher than in the subsoil for both variables.

Our study provides strong experimental evidence for viscosity-controlled diel DOC fluctuations in the Weierbach. Previous work by Schwab et al. (2016) in the Weierbach catchment had shown that an increase in riparian water temperature during the day led to a decrease in riparian water viscosity and subsequently to an increase in hydraulic conductivity. This viscosity effect resulted in an increased inflow of riparian groundwater to the stream in the afternoon – from the topsoil of the riparian zone to the creek. The timing of the daily minima of viscosity in the afternoon is consistent with the timing of the daily

maxima of DOC and SUVA-254. Besides the timing of the viscosity minima, the high-frequency anomalies provide another solid indication that the viscosity effect triggers an increased inflow of terrestrial DOC to the creek in the afternoon. The strong regression between the viscosity and the SUVA-254 anomalies and especially the regression between the viscosity and the DOC anomalies showed that viscosity, SUVA-254 and DOC had very similar diel dynamics.

The regressions between the discharge anomalies and the anomalies of viscosity, SUVA-254 and DOC resulted in different

slope directions and values depending on the season. This behavior can be explained by the existence of two different opposing processes that are controlling the diel discharge fluctuations: the viscosity effect during the dormant season and evapotranspiration during the growing season. We argue that different spatial impacts are the reason, why during the growing season evapotranspiration is controlling discharge but the viscosity effect is controlling the DOC concentrations and the SUVA values in the stream. While the viscosity effect is only present in the topsoil of the riparian zone, the plants

transpire water from deeper soil depths (Bond et al., 2002; Schwab et al., 2016). Especially the upper parts of the soil had high SUVA-254 and DOC concentration values.

Different models for simulating autochthonous DOC dynamics exist (Fasching et al., 2016; Worrall and Moody, 2014). However, these models are partly contradictory and no state-of-the-art model has been established so far. In addition, we did not have all the data required to run these models. Consequently, we did not simulate the autochthonous DOC dynamics.

However, we developed a perceptual model to explain the observed diel DOC and SUVA-254 anomalies, depending on instream processes and terrestrial input (Figure 6). The conceptual model follows the main results of Fasching et al. (2016), stipulating that the instream DOC production is higher with increasing stream water temperature and increasing photosynthetically active radiation (PAR). With the perceptual model that is illustrated in Figure 6, we can also explain the observed smaller slopes resulting from the regression between the SUVA-254 and the DOC anomalies during the growing

season. The amplitudes of the diel DOC anomalies stayed relatively constant over the whole year, while the diel amplitudes of SUVA-254 decreased during the growing season. We argue that an increasing importance of instream processes during the growing season leads to a decrease in SUVA-254.





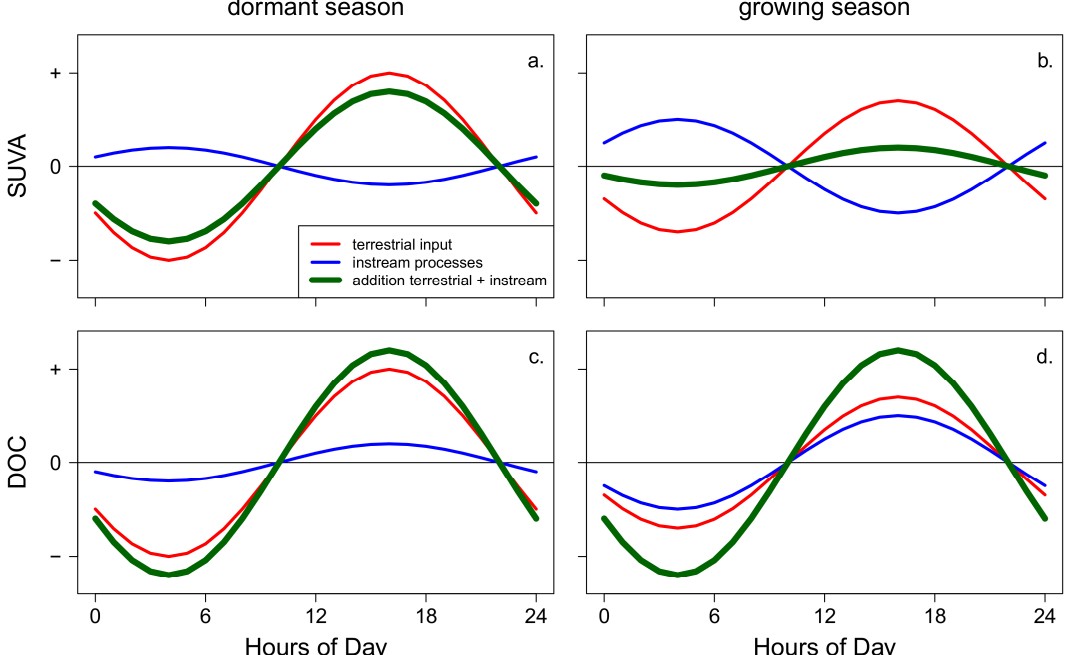

**Figure 6: Perceptual understanding on the diel SUVA and DOC fluctuations and its dependence on instream processes or terrestrial input and the resulting superposition of both processes.**

In our perceptual model (Figure 6), the diel SUVA-254 fluctuations resulting from instream processes show an opposite

pattern compared to the diel SUVA-254 fluctuations resulting from terrestrial DOC input. This can be explained by differences in the aromaticity of DOC of the two processes. Depending on the magnitude of both processes, the resulting superposition of both processes may change the diel pattern or not. As a consequence of the increasing stream water temperature and PAR in summer, SUVA-254 fluctuations resulting from instream processes are much higher during the growing season than during the dormant season (Figure 6 a,b) (Fasching et al., 2016). On the other side, the diel SUVA-254

fluctuations resulting from terrestrial DOC input triggered by viscosity effects are smaller during the growing season due to a decrease of the viscosity fluctuations in summer (Schwab et al., 2016). By overlaying the instream and the terrestrial effect on SUVA-254, the resulting diel SUVA-254 fluctuations are higher in the dormant season than in the growing season.

Contrarily to the SUVA-254 fluctuations, the diel DOC fluctuations resulting from instream processes and terrestrial input are in phase. They have their maxima in the afternoon when the stream water temperature and the PAR (influencing the

15 instream processes) are at their maxima and the riparian water viscosity (influencing the terrestrial input) has its minima. During the growing season (Figure 6d), the diel DOC fluctuations induced by instream processes are higher than during the




dormant season and the DOC fluctuations resulting from terrestrial input are smaller (smaller viscosity fluctuations) than during the dormant season (Figure 6c). Consequently, the overlaying of both effects results in similar DOC fluctuations during the growing and the dormant seasons (Figure 6c,d). In other catchments the relative proportion of the different processes is probably different, resulting in other overall diel fluctuations.

In addition to the diel fluctuations, we observed a seasonal pattern in the daily mean values of SUVA-254 and DOC concentrations. In the Weierbach creek we observed higher SUVA-254 values and DOC concentrations during the low flow periods compared to high flow periods, while Lee et al. (2016) and Fasching et al. (2016) described lower SUVA-254 values during dry, respectively baseflow conditions (Figure 2). This could likely be explained by different flow paths of the water contributing to stream flow. During summer low flow, we suspect that only a few source areas in the riparian zone contribute

to streamflow. Those riparian source areas have higher SUVA-254 values and DOC concentrations (Figure 5). During periods with higher discharge, especially in winter and early spring, a dilution effect leads to decreasing SUVA-254 values and DOC concentrations. Larger areas with lower SUVA-254 values and DOC concentrations contribute to streamflow. During those wet conditions, subsurface flow, whose SUVA-254 and DOC signature is represented by the shallow groundwater end-member (Figure 5), generated a large part of the discharge.

## 15   5 Conclusion

We observed diel DOC fluctuations in the Weierbach catchment over a complete year during periods that were not affected by rainfall-runoff processes. By means of the SUVA-254 index, serving as an indicator for DOC aromaticity, we found that an increased input of DOC with terrestrial origin was responsible for the peak in DOC concentrations in the afternoon. Higher SUVA-254 values indicate a higher aromaticity of DOC and therefore an increase of DOC from terrestrial

(allochthonous) sources. We could explain the increased input of terrestrial DOC in the afternoon with the viscosity effect. Water passing the riparian zone before entering the creek is heated in the riparian zone during the day. Warmer water has a decreased viscosity and therefore the hydraulic conductivity increases. Consequently, more water from near surface zones that are rich in terrestrial DOC is entering the creek in the afternoon. Our study described a new process that can explain diel DOC fluctuations in streams. We argue that the analysis of diel DOC fluctuations should not only focus on instream

processes, but also on surface areas in the vicinity of the creek. Moreover, viscosity driven diel hydrological flow processes have to be taken into account for understanding diel DOC dynamics in streams.

For further studies, we suggest to combine the UV-Vis spectrometer measurements with fluorescence spectrometry measurements to gain even more detailed information about the origin of the DOC. Furthermore, a more detailed insight into the instream DOC processes would be an interesting aspect of future research. Additionally, we hope that our study could

raise the awareness that viscosity driven input of terrestrial DOC can explain diel DOC fluctuations in stream water. We believe that this effect can be also detected in other catchments, but depends on the catchment-specific interplay of both interacting processes.



**Acknowledgements**

We acknowledge the FNR (Fonds National de la Recherche Luxembourg) for having funded this research through the AFR PhD grant (Grant 6931545). Additional funding was provided through the FNR-DFG CAOS-2 project (INTER/DFG/14/02) and the DFG funded CAOS project (FOR 1598). We also thank Jean François Iffly, François Barnich and Jérôme Juilleret

for their support during field activities and in the laboratory. A special thanks goes to Christophe Hissler, who provided us with the biweekly end-member dataset that was acquired during the project FNR/CORE/SOWAT (C10/SR/799842).

**Competing interests**

The authors declare that they have no conflict of interest.

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
