# Peer review of "Diel fluctuations of viscosity-driven riparian inflow affect streamflow DOC concentration"

_Biogeosciences, 2017_

## Referee Comment (RC1) · Anonymous Referee #1 · 17 Jul 2017

General comments:

The paper analyses diel concentration dynamics of DOC in a headwater stream. The analysis explicitly addresses dry conditions and thus focuses on the internal controls of the system rather than external hydrological drivers such as snowmelt and rainfall. I think it has an important and general value to better understand stream concentrations dynamics and moreover to disentangle responses of external forcing and internal controls.

In this paper the authors argue that diel DOC concentration variability is driven by temperature-induced changes of hydraulic conductivity (via water viscosity) which leads to changes in the magnitude of groundwater discharge and the associated discharge of DOC and ultimately to variations of instream concentrations. Although the

authors claim that this is the only mechanism that can explain the observed pattern, they do not provide enough evidence. I would like to challenge the interpretation by asking to perform a mixing analysis. Assume that all variations in discharge are driven by viscosity-induced discharge of groundwater (or water from the riparian zone if you like) what would be the DOC concentration in this water that explains the observed DOC concentrations in the stream. Do these concentrations match the observed DOC concentrations in the riparian zone? This would be a simple test to check the plausibility of the proposed mechanism. Moreover one could have a look at the concentration-discharge relationships: From Figure 1 c and d I see a dilution pattern in DOC concentrations at multi-day timescales. Low discharge=High concentration which means that the load (C*Q) remains broadly constant, indicating a constant source. This relationship does not hold for the diurnal discharge peaks, here we see increasing concentrations with increasing discharge - suggesting a temporarily dynamic source, potentially viscosity effects. In g and h this seems reversed. DOC follows discharge at the multi-day time scale- meaning a drastic reduction of DOC load, while daily DOC peaks occur at daily discharge minima. This last fact largely counteracts the "viscosity hypothesis". Thus for ∼50% of the data the viscosity hypothesis must be rejected. I encourage the authors to "harvest" their data in multiple ways to support their hypothesis or better to explain the multiple processes rather than focusing on a one-sided interpretation.

Given the fact that the study was conducted in a potentially well studied catchment, I wonder why the data is so sparse. The entire line of argumentation is based on a single temperature observation location which is used to explain catchment scale effects. I think this "extrapolation" is not justified. Moreover, the authors claim that at this location, there is high subsurface flow. However, they do not provide any data to support this. How do I know that this location is representative for the entire catchment? I strongly recommend better explaining the hydrologic aspects of how this catchment functions. I mostly missed some hydraulic gradients between the stream and the groundwater, some numbers on the amount of GW discharge and its spatial distribution and how temperatures, particularly streambed temperatures vary spatially.

I am aware of the Schwab et al. 2016 study but also there, these essential information are not provided.

My main technical concern is the position of the temperature sensor. If I understand correctly temperatures are measured at 10 cm below the land surface in the unsaturated zone. I wonder how this temperature can be representative for the water that is discharging into the stream. In the unsaturated zone water flows vertically, driven by gravity. Thus more or less horizontal flow, discharging into the stream is bound to saturated, Darcian flow. Moreover the relationship between hydraulic conductivity and viscosity is for saturated conditions. Under variably saturated conditions, saturation should have a much larger influence. No data on water saturation is reported. Anyway I doubt that the unsaturated zone is the source for the stream water. Thus to evaluate the effects of viscosity temperatures should be measured at groundwater discharge locations directly in the streambed.

In summary, this manuscript presents only a modest amount of (spatial) data to support the "viscosity hypothesis". My impression is that the interpretation of the data is one-sided towards this hypothesis. I encourage the authors to acknowledge the pattern which are obviously in their data and provide an analysis that is accounting for the different controls of DOC concentration which vary between seasons.

Specific comments:

I find "riparian water" is a misleading term - hydrologically there is no difference between soil and riparian water - both are in the unsaturated zone. So at least it should be defined what exactly is meant here.

P.3. 15-20: This is exactly the point where the authors are on the wrong track. Water (groundwater - saturated zone) cannot flow through the unsaturated zone ( I guess that the riparian zone is unsaturated because of 1) p5.l.3 sampling of riparian water was with suction cups, and 2) sampling depth is 10 cm below the ground) and then entering the stream. I encourage the authors to provide a conceptual model on the
water fluxes and heat transport at the site

P.4.l.14-15: Please provide a reference, better data confirming that this location has high GW inflow. Moreover the sampling and measuring locations should be provided, in a way that the reader knows if the riparian water was sampled in 10 cm, 1m or 10m distance from the stream. Also: where are the GW wells? I think the spatial relationships are important. Please provide this in a map or a cross-section.

p.5 l.24.: I don't understand what is meant by anomaly. This seems important for the further analysis but I don't get it. Is it a time shift between the variables? Or is it the difference between the 24h moving average time series and the original time series? I guess the latter. If so, what has been done is a simple form of spectral high pass filtering. You cancel out the low frequencies and only keep high frequencies of 1/24 dˆ-1. This should be better explained, best in terms of common time series analysis terminology.

p.5.l.25-26. If a 24h moving average is applied you filter all fluctuations with shorter timescales.

p.5. l 29.-33. Are periods without DOC fluctuation also periods without temperature variation? if so this would support the viscosity hypothesis. Please report temperature and viscosity fluctuations in these periods as well.

p.10. l.15: What I see in Figure 1 is that for all times SUVA and DOC are highly correlated - also the minima. So far as I can see there is no indication that SUVA is particularly high when DOC is high. This is also supported by the good correlation between SUVA and DOC fluctuation in Fig. 3. Thus SUVA seems a good indicator for DOC concentration and thus not only the maxima, but generally SUVA indicates inputs not changes in DOC quality at this site.

p.11.l5: Here I would disagree, the evidence is not strong.

p.11. l.14-20: I think the reversed relationship between concentration and discharge is

really striking and is not explained. e.g. p.l.17 "different spatial impacts" what is this exactly, how can you assess this by having only measured at a single location. I think if the authors could figure out how the controls of DOC concentration change over the season because the importance of different controlling factors vary, would make this work a strong contribution.

p.12.l4 ff: I think this perceptual model should be extended by discharge effects. The authors should remember that their main line of argumentation is the increase discharge of water driven by viscosity. Comparing Fig.1 g and h with 6 d I would not bet that DOC inputs are high in the afternoon, concentration is high, but discharge is low. So again, also consider loads, not concentrations alone.

Figure 1: Please provide temperature data as well.

Figure 3: This is a tough one for ∼8% of male population! Anyway, in 3d the green regression line does not match the data well - visually it should be steeper.

[Figure]

---

## Referee Comment (RC2) · Anonymous Referee #2 · 21 Sep 2017

The study deals with an original time scale of dissolved organic matter variations as diel cycles have been less studied than event responses or seasonal patterns and long-term trends. DOM is described via 2 parameters: DOC concentration and its properties through SUVA index as a proxy of the aromaticity. The studied hypothesis is also original and mainly supported by a previous work by Schwab et al in 2016 that compared the respective role of evapotranspiration and riparian inflow changes (due to temperature-driven viscosity changes) on diel fluctuations of stream flow. For riparian GW temperature, DOC stream concentration and corresponding SUVA-254, diel cycles are in phase over the whole time series with daily max occurring in the afternoon (between 2pm and 6 pm). Amplitude of the cycles is minimal in winter. Amplitude of riparian shallow groundwater and DOC concentration cycles is relatively constant dur-

ing the rest of the period. Amplitude of SUVA-254 cycles is high in spring (at the end of the dormant period) and small in summer (middle of the growing period). For discharge, diel cycles change in phase between the dormant season (morning max) and the growing season (afternoon max) and disappear during winter (start of the dormant period). Amplitude of the discharge diel cycles seems higher in spring (end of dormant period) than in the growing period. From these observations the authors suggest that the variations of riparian flow (due to water viscosity variations with the temperature) are the major control of DOM diel cycles. I found the manuscript well written with clear messages. The analyses and supporting data set are valuable. However I found the conclusion on the respective hypothesized controlling processes (in-stream biology versus riparian flow conductivity) quite hasty and maybe too categorical regarding what is effectively observed and demonstrated. The results support the hypothesis but some questions remain and interpretation should remain more careful in my point of view (please see specific comments).

Detailed comments and questions

Introduction

p.2 line 3: Do you have an idea of the relative concentration levels of DOC and DIC in the study stream?

p.5 line 5 indeed photodegradation has been shown significant on highly brown DOM coming from peatlands (references cited by the authors). I am not sure that it has been reported as important on forested-derived DOM

Methods

p.4 lines 5-6 This point may be an output from previous research conducted on this well-studied catchment but how the significance of riparian zone contribution has been demonstrated? And quantifications if available would be useful

p. 4 lines 11-13 provide information about average annual pattern of flow. Similar

information about the annual behaviour of DOC concentration and SUVA-254 would be useful to understand the catchment: from Fig. 2 it seems that mean DOC and mean SUVA-254 are maximal in summer low-flow period (increase from Feb to Aug 2014 and from Feb to June 2015). If riparian subsurface flow is the main source of aromatic DOC, I expected this contribution being higher in high flow periods and lower in low flow periods when catchment saturation decreases and therefore minimal DOC and SUVA values in this low flow period. . .

p. 4 lines 16-17 I think there is an error: unless I am mistaken a variation of 5°C leads to a viscosity change of 2% only. Using Eq. 3 from Schwab et al. (2016):

$\eta(T°)=e^{(3.7188+578.919/(137.546+T°))}$

If T° is the temperature in Kelvin degree (as said "T in K"). I think that the 12 to 15% of change of viscosity have been deduced by applying the formula with Temp in Celsius degree, right?

As brief example, for T=15 Celsius deg (288 Kelvin deg) I found Nu=164 Pa.s

for T=20 Celsius deg (293 Kelvin deg) the formula gives Nu=161 = 164 -2%

p. 4 lines26-31: Did you compare also the absorbance values at 254 nm from spec-tro::lyser and from the lab? (Since Absorbance values are available for the end-members - p4 lines 33-34- I suppose that some exist for the stream as well. . .?)

p. 5 lines 1-5 sampling points for the end-members, as well as the stream station and the temperature monitoring point should be located on a map of the site.

p. 5 line 3 Regarding the method to sample riparian water, I wonder if the riparian area was effectively fully saturated? On another hand, viscosity of riparian water is calcu-lated from a temperature sensor located in the riparian groundwater at 10 cm depth so I imagine that riparian groundwater remains shallower than those 10 cm depth?. . . Is the water table level in this specific zone monitored (that could help giving the local hydraulic gradient with the stream)? Is riparian water sampled at 6 different depths too

or only at 10 cm? Do you observe any vertical variability of DOC and SUVA in this riparian zone as shown for soil water in Fig 5 b,d? Regarding Fig 5a, the DOC richness is finally much closer between riparian water and groundwater and low.

p.5 line 13 It is not clear in the following which analyses do use this smoothed SUVA time series (obtained from 3 hours moving window) or the raw time series: p. 5 line 25 "the original time series with the 15 min time intervals" is used to compute the distance to daily average

Results

p.7 The difference in amplitude of diel discharge cycle between dormant and growing period is not characterized but amplitude of the discharge diel cycles seems higher in spring (end of dormant period) than in the growing period? Maybe a scale effect to due difference in base flow?

p. 7 Figure 2: It would help to represent dormant and growing periods on the graph by color or shadings or vertical lines for instance

Figure 5: If possible with the scales, the corresponding values in stream water could be added in (a) and (c) to have in mind the relative position of stream between the end members

Discussion

p. 10 lines 5-9: Correlation between DOC and SUVA_254 fluctuations sounds consistent and I certainly agree with the authors on the value of SUVA_254 or such indices as proxy of DOM composition and properties. However, there is a point absolutely not discussed here: the fact that DOC concentration value is computed by the spectro::lyser algorithm using the absorbance value at 254 (or absorbance at 252 & 255nm). Other absorbance values are obviously included in this concentration estimate but the DOC and SUVA variables used here are somehow both functions of measured absorbance at 254 nm, so that their correlation is not fully surprising. . .At least I feel that it deserves

a word in the discussion. See also my comment on (p. 4 lines26-31)

p. 10 lines 14-15 I feel the rejection of the first hypothesis arrives a little bit fast. The absence of in-stream processes is not fully demonstrated to my opinion. Microbial processes are numerous, here you assume DOC concentration increase due to in-stream production should exhibit a low aromaticity and therefore a low SUVA but i) biological processes that recycle the DOC are numerous enough to lead to complex antagonistic results; ii) keep in mind that SUVA is only a proxy of the complex composition of DOC; iii) and again that is this case SUVA is computed using absorbance properties only .

On the other hand, all the conclusions are based on relationships between DOC, SUVA and viscosity which is actually an interpretation of measured temperature variations. Therefore, what is established strictly speaking is that DOC and SUVA variations are correlated with temperature in riparian water isn't it? I wonder if the correlations would have been poorer using for instance stream temperature? And temperature is a factor control of viscosity but also many processes, biological processes, evapotranspiration. . .

If I didn't make a mistake on comment regarding (p. 4 lines 16-17) above, variations of $5°C$ would induce a change (in viscosity and thus also) in hydraulic conductivity of 2%, which is a very small change, and even if the 10-15% of variations are right, I wonder how significant it is on the flow from this area. If you had an estimate of the range of hydraulic conductivity and of the hydraulic gradient to stream (via measurement of groundwater level) this would help to understand the relative weight of such an increase of the viscosity?

Finally there are still missing pieces of discussion:

(p. 8 & p. 9 lines 1-2) Correlation between DOC and SUVA daily variations are stronger during dormant period: why if their diel fluctuations have the same origin (riparian flow)? Would this be related to a change of riparian DOM composition? If so, such change would be visible on the end-members samples? Looking back at Fig 6, it

appears to me that this difference is explained actually by stronger in-stream processes that would have during growing season comparable effects to viscosity fluctuations. So that seasonal processes would be also a dominant control, isn't it?

p. 9 lines 10-11 If riparian water is responsible for diel increase of DOC stream concentration, I found it surprising that the DOC concentration in the riparian water is finally rather low compared to soil water in the hillslope...

p. 11 lines 1-4 see my suggestion for Figure 5

p. 11 lines 14-20 Schwab et al. (2016) concluded that Q fluctuations during dormant season was indeed resulting from viscosity changes resulting in variable riparian flow to stream, but in the growing season , the role of evapotranspiration fluctuations was dominant (leading to diel cycle inversion). The authors explain the fact that Q and DOC are not affected by the same processes because of the relative influence of those processes on the respective "sources" of water and DOC. ET controls Q cycle affecting the whole catchment storage while viscosity controls the DOC from riparian upper layers. So at the end, those stream signatures are integrating various catchment processes and disentangling those processes faces the same issue as distinguishing the processes that can control seasonal cycles on water quality. Maybe in further studies, it would be worth to try looking at some other parameters that could play the role of riparian flow tracers to support further the hypothesis.

p. 13 lines 5-14: this answers partially my comment on (p. 4 lines 11-13). However I found this seasonal pattern quite surprising. In many study, Hillslope subsurface flows merely active during wet conditions intercept the riparian area flushing somehow their upper soil layers rich in DOC leading to high DOC concentration (and more aromatic as well). During low flow, saturated area extension is decreased, and connection between those DOC sources and the stream can be interrupted. Flow is sustained mainly by groundwater which is poor in DOC so should lead to minimal DOC concentrations excepted if autochthonous production increases this DOC concentration. The proposed

interpretation for Weierbach catchment should be discussed regarding general understanding that have been proposed elsewhere.

It would be interesting to have an idea of the importance of the variations we are looking at (as percentage of flow/concentration/SUVA mean value). I do not discuss the interest of the topic that has been scarcely studied so far but I think that keeping in mind the relative orders of magnitude of the studied phenomena sounds relevant

Conclusion

p. 13 lines 26-29: I wonder if other tracers unrelated to carbon dynamics could be interesting for tracking independently the riparian flow for instance. I would also suggest the use of O2 probes to try catching indirect information on metabolic activity of the stream?

---

## Author Comment (AC1) · 1 Nov 2017

General comments:

The paper analyses diel concentration dynamics of DOC in a headwater stream. The analysis explicitly addresses dry conditions and thus focuses on the internal controls of the system rather than external hydrological drivers such as snowmelt and rainfall. I think it has an important and general value to better understand stream concentrations dynamics and moreover to disentangle responses of external forcing and internal controls.

In this paper the authors argue that diel DOC concentration variability is driven by temperature-induced changes of hydraulic conductivity (via water viscosity) which leads to changes in the magnitude of groundwater discharge and the associated discharge of DOC and ultimately to variations of instream concentrations. Although the authors claim that this is the only mechanism that can explain the observed pattern, they do not provide enough evidence. I would like to challenge the interpretation by asking to perform a mixing analysis. Assume that all variations in discharge are driven by viscosity-induced discharge of groundwater (or water from the riparian zone if you like) what would be the DOC concentration in this water that explains the observed DOC concentrations in the stream. Do these concentrations match the observed DOC concentrations in the riparian zone? This would be a simple test to check the plausibility of the proposed mechanism. Moreover one could have a look at the concentrationdischarge relationships: From Figure 1 c and d I see a dilution pattern in DOC concentrations at multi-day timescales. Low discharge=High concentration which means that the load (C*Q) remains broadly constant, indicating a constant source.
This relationship
does not hold for the diurnal discharge peaks, here we see increasing concentrations with increasing discharge - suggesting a temporarily dynamic source, potentially viscosity effects.

We realize that we have to clarify some important things in the manuscript to avoid misunderstandings: the time scale of interest and our process understanding of the Weierbach catchment in general.
In this manuscript, we only focus on diel fluctuations and not on the event, multiday, seasonal or annual time scales. The viscosity effect is only of importance for variations on a daily time scale and we do not attempt to explain differences in DOC concentrations on longer time scales or between

seasons. We do not assume that all variations in discharge are driven by viscosity-induced discharge of groundwater, but only the diel variations of discharge and DOC under very specific conditions. We will clarify this in the revised manuscript. Hence, we do not believe, that a mixing analysis can improve our process understanding on a daily time scale.

We addressed the event and seasonal patterns of DOC in another manuscript that is currently under review and that focuses on the event and annual time scale (Schwab et al., 2017). After reaching a storage threshold during wet conditions (and therefore during high flows) an additional runoff process is playing an important role: subsurface flow / shallow groundwater flow with low DOC and SUVA-254 values. This leads to decreasing DOC and SUVA254 values with increasing discharge and vice versa (Figure 1 b,c,d; Figure 2 b,c,d).

In g and h this seems reversed. DOC follows discharge at the multiday time scale- meaning a drastic reduction of DOC load, while daily DOC peaks occur at daily discharge minima. This last fact largely counteracts the "viscosity hypothesis". Thus for _50% of the data the viscosity hypothesis must be rejected. I encourage the authors to "harvest" their data in multiple ways to support their hypothesis or better to explain the multiple processes rather than focusing on a one-sided interpretation. Given the fact that the study was conducted in a potentially well studied catchment,

Figure 1 g,h is not a good example for the multiday relationship between DOC and discharge; it is not representative for the multiday timescale. Yet DOC does not follow discharge during the growing season (see Figure 2; daily mean values of DOC and discharge). Figure 1 is focusing on the diel fluctuations.

The discharge minima in the afternoon during the growing season are triggered by evapotranspiration. This does not exclude the presence of the visoscosity effect. Most likely, the viscosity effect is only hidden by the effect of evapotranspiration. This was already discussed in (Schwab et al., 2016).

The SUVA254 peak occurs always in the afternoon, both during the growing and the dormant season. This indicates that an increasing amount of terrestrial DOC is entering the stream in the afternoon. And this correlates well with the viscosity effect.

I wonder why the data is so sparse. The entire line of argumentation is based on a single temperature observation location which is used to explain catchment scale effects. I think this "extrapolation" is not justified. Moreover, the authors claim that at this location, there is high subsurface flow. However, they do not provide any data to

support this. How do I know that this location is representative for the entire catchment?

Indeed, we have to explain that better in this manuscript. We observed the temperature fluctuations at different locations. However, we only had one spot, where we had high-frequency data for the entire observation period and therefore selected this observation.

We will revise this in the revision and describe the locations and the other observations to show that this observation is representative for water temperature fluctuation in the riparian zone.

I strongly recommend better explaining the hydrologic aspects of how this catchment functions. I mostly missed some hydraulic gradients between the stream and the groundwater, some numbers on the amount of GW discharge and its spatial distribution and how temperatures, particularly streambed temperatures vary spatially. I am aware of the Schwab et al. 2016 study but also there, these essential information are not provided.

We will better describe the hydrologic aspects of the catchment in the revised manuscript. The riparian zone extends 1 to 5 meter from the stream and is around 1m deep. The stream flows on solid, impermeable, unweathered bedrock. Hence, most of the groundwater is flowing through the riparian zone into the stream (hydraulic gradients from the riparian groundwater to the stream).

My main technical concern is the position of the temperature sensor. If I understand correctly temperatures are measured at 10 cm below the land surface in the unsaturated zone. I wonder how this temperature can be representative for the water that is discharging into the stream. In the unsaturated zone water flows vertically, driven by gravity. Thus more or less horizontal flow, discharging into the stream is bound to saturated, Darcian flow. Moreover the relationship between hydraulic conductivity and viscosity is for saturated conditions. Under variably saturated conditions, saturation should have a much larger influence. No data on water saturation is reported. Anyway I doubt that the unsaturated zone is the source for the stream water. Thus to evaluate the effects of viscosity temperatures should be measured at groundwater discharge locations directly in the streambed.

We consider the riparian zone as being saturated. At least the parts with high inflow to the stream are saturated during most parts of the year. At the location of the sensor the soil is saturated during

the whole observation period and most of the inflow must enter the stream through the riparian zone.

Review #1 clearly points to our limited description of the catchment in the current manuscript. This needs further clarification in the revised manuscript and the specific site characteristics of the Weierbach catchment will be better discussed.

In summary, this manuscript presents only a modest amount of (spatial) data to support the "viscosity hypothesis". My impression is that the interpretation of the data is onesided towards this hypothesis. I encourage the authors to acknowledge the pattern which are obviously in their data and provide an analysis that is accounting for the different controls of DOC concentration which vary between seasons.

As already mentioned before, this paper explicitly focuses on the explanation of diel DOC fluctuations and not on the seasonal pattern of DOC. The seasonal patterns cannot explain the diel DOC pattern that we observed in the stream.

What we clearly see in our observations is the fact that the diel SUVA254 maxima are in the afternoon, both during the growing and the dormant season. An elevated SUVA254 value indicated an increased amount of aromatic/terrestrial DOC in the stream. This increased amount of DOC likely comes from the riparian zone, where the viscosity effect takes place. In general, science develops by testing alternative hypothesis (viscosity effects) and to show evidence for this hypothesis. As we can only reject hypothesis, but not completely proof, we can only show the indications we have and mention the other hypothesis which we rejected.

Specific comments:

I find "riparian water" is a misleading term - hydrologically there is no difference between soil and riparian water - both are in the unsaturated zone. So at least it should be defined what exactly is meant here.

We will define this more precise in the revised manuscript. Yet we disagree. The riparian zone can be saturated and unsaturated. We could either use the term saturated near stream areas or saturated riparian areas.

P.3. 15-20: This is exactly the point where the authors are on the wrong track. Water (groundwater - saturated zone) cannot flow through the unsaturated zone ( I guess that the riparian zone is unsaturated because of 1) p5.l.3 sampling of riparian water

was with suction cups, and 2) sampling depth is 10 cm below the ground) and then

entering the stream. I encourage the authors to provide a conceptual model on the

water fluxes and heat transport at the site

Our sampling location in the riparian zone is always saturated up to the soil surface. It is possible to sample with suction cups in the saturated zone – the suction cups were installed in the beginning to allow sampling under saturated and unsaturated conditions in the whole catchments – this location, however is saturated throughout the year. We will state this clearer in the revised manuscript.

P.4.l.14-15: Please provide a reference, better data confirming that this location has

high GW inflow.

The flow from the riparian zone to the stream is continuously monitored by thermal cameras and the contribution from different sections of the riparian zone to discharge is measured by dense discharge measurements (salt dilution method) along the stream. This is still work in progress as part of two PhD projects.  We have to admit that the statement ("location of high GW inflow") is somewhat vague. We know that the hydraulic gradient is towards the stream and that the location is constantly saturated.

Moreover the sampling and measuring locations should be provided,

in a way that the reader knows if the riparian water was sampled in 10 cm, 1m or

10m distance from the stream. Also: where are the GW wells? I think the spatial

relationships are important. Please provide this in a map or a cross-section.

We will include a map with the sampling and measurement locations. Maps with the sampling and measurement locations are already published/under review in (Schwab et al., 2016, 2017).

p.5 l.24.: I don't understand what is meant by anomaly. This seems important for the

further analysis but I don't get it. Is it a time shift between the variables? Or is it the

difference between the 24h moving average time series and the original time series?

I guess the latter. If so, what has been done is a simple form of spectral high pass

filtering. You cancel out the low frequencies and only keep high frequencies of 1/24

dˆ-1. This should be better explained, best in terms of common time series analysis

terminology.

Yes, it is the difference between the 24h moving average time series and the original time series. We will revise the explanation.

p.5.l.25-26. If a 24h moving average is applied you filter all fluctuations with shorter

timescales.

We do not have significant fluctuations shorter than a daily timescale. In this manuscript, we want to understand the diel fluctuations. Hence, we applied a 24h moving average.

p.5. l 29.-33. Are periods without DOC fluctuation also periods without temperature variation? if so this would support the viscosity hypothesis. Please report temperature and viscosity fluctuations in these periods as well.

The data that we present in our manuscript includes only days that are not influenced by rainfall-runoff processes. Outside rainfall-runoff processes, we observed only minor temperature fluctuations during days with small DOC fluctuations. Days without DOC fluctuations are normally influenced by rainfall-runoff processes, where the diel temperature fluctuation is also disturbed or not existing.

p.10. l.15: What I see in Figure 1 is that for all times SUVA and DOC are highly correlated - also the minima. So far as I can see there is no indication that SUVA is particularly high when DOC is high. This is also supported by the good correlation between SUVA and DOC fluctuation in Fig. 3. Thus SUVA seems a good indicator for DOC concentration and thus not only the maxima, but generally SUVA indicates inputs not changes in DOC quality at this site.

As the reviewer mentioned, there is a correlation between absorbance at 254nm and DOC. According to the measurement method of the spectrometer, DOC is calculated based on absorbance 254nm. However DOC is not only calculated based on the absorbance at 254nm but also based on the absorbencies at other wavelengths. SUVA 254 is calculated as the absorbance at 254nm normalized by the DOC concentration (SUVA254 = A254/DOC). Consequently, an increase in SUVA254 is based on an increase in A254 that is larger than the increase in DOC concentration. Therefore on increase in SUVA254 is not (only) based on an increase of DOC in general but on an increase in more aromatic DOC.

We will include this information into the discussion to clarify this point.

SUVA254 is indicator for the quality changes of DOC. The quality changes of DOC can be affected by terrestrial input and instream processes.

p.11.l5: Here I would disagree, the evidence is not strong.

We will change the wording in indicate. We think that there is a strong indication due to…...

p.11. l.14-20: I think the reversed relationship between concentration and discharge is

really striking and is not explained. e.g. p.l.17 "different spatial impacts" what is this exactly, how can you assess this by having only measured at a single location. I think if the authors could figure out how the controls of DOC concentration change over the season because the importance of different controlling factors vary, would make this work a strong contribution.

At first, this work is not about seasonal controls on DOC concentration, but solely explaining the diurnal pattern of DOC and SUVA. The reversed relationship between discharge and viscosity is explained in previous work (Schwab et al., 2016). The viscosity effect (dominant factor controlling discharge fluctuations during the dormant season) has an impact only on the upper part of the saturated riparian zone. Evapotranspiration (dominant factor controlling discharge fluctuations during the growing season) has also on impact on water in deeper layers. The different timing of the diel discharge extrema between growing and dormant season comes from the seasonally changing importance of evapotranspiration and viscosity (Schwab et al., 2016). Nevertheless, we could show in Schwab et al. (2016) that the viscosity effect is always present. As it has only an impact on the upper layer that is richer in DOC, it creates DOC maxima in the afternoon throughout the year. The SUVA254 maxima in the afternoon indicates that the increased DOC input is from terrestrial sources. Hence, this is a strong indication that the increased input comes from near surface layers (with increased SUVA254 and DOC values) where the viscosity effect has an impact.

p.12.l4 ff: I think this perceptual model should be extended by discharge effects. The authors should remember that their main line of argumentation is the increase discharge of water driven by viscosity. Comparing Fig.1 g and h with 6 d I would not bet that DOC inputs are high in the afternoon, concentration is high, but discharge is low. So again, also consider loads, not concentrations alone.

As already mentioned above, the rainfall-runoff responses are not affecting the diel signal. We will carefully revise the manuscript so that it becomes ultimately clear that this study is solely aiming at the diel pattern of DOC and related SUVA. We do not aim at distinguishing seasonal controls on DOC here, which are clearly related to hydrological processes and rainfall-runoff responses. The behavior of DOC on event and seasonal time scale is described in another manuscript (Schwab et al., 2017).

Figure 1: Please provide temperature data as well.

Temperature has the same pattern as viscosity, as viscosity is a function of temperature. Therefore, we decided not to include temperature.

Figure 3: This is a tough one for _8% of male population! Anyway, in 3d the green

regression line does not match the data well - visually it should be steeper.

We will change the colors.

We checked the regression again, and it is the proper regression using least square fit. There are so many data points (especially in the center) that the visual impression can be misleading. The fewer data points outside the center can be less important for the regression.

**Citations**

Schwab, M., Klaus, J., Pfister, L. and Weiler, M.: Diel discharge cycles explained through viscosity fluctuations in riparian inflow, Water Resour. Res., 52(11), 8744–8755, doi:10.1002/2016WR018626, 2016.

Schwab, M. P., Klaus, J., Pfister, L. and Weiler, M.: How runoff components affect the export of DOC and nitrate: a long-term and high-frequency analysis, Hydrol. Earth Syst. Sci. Discuss., 1–21, doi:10.5194/hess-2017-416, 2017.

---

## Author Comment (AC2) · 1 Nov 2017

The study deals with an original time scale of dissolved organic matter variations as diel cycles have been less studied than event responses or seasonal patterns and long-term trends. DOM is described via 2 parameters: DOC concentration and its properties through SUVA index as a proxy of the aromaticity. The studied hypothesis is also original and mainly supported by a previous work by Schwab et al in 2016 that compared the respective role of evapotranspiration and riparian inflow changes (due to temperature-driven viscosity changes) on diel fluctuations of stream flow. For riparian GW temperature, DOC stream concentration and corresponding SUVA-254, diel cycles are in phase over the whole time series with daily max occurring in the afternoon (between 2pm and 6 pm). Amplitude of the cycles is minimal in winter. Amplitude of riparian shallow groundwater and DOC concentration cycles is relatively constant dur-ing the rest of the period. Amplitude of SUVA-254 cycles is high in spring (at the end of the dormant period) and small in summer (middle of the growing period). For discharge, diel cycles change in phase between the dormant season (morning max) and the growing season (afternoon max) and disappear during winter (start of the dormant period). Amplitude of the discharge diel cycles seems higher in spring (end of dormant period) than in the growing period. From these observations the authors suggest that the variations of riparian flow (due to water viscosity variations with the temperature) are the major control of DOM diel cycles. I found the manuscript well written with clear messages. The analyses and supporting data set are valuable. However I found the conclusion on the respective hypothesized controlling processes (in-stream biology versus riparian flow conductivity) quite hasty and maybe too categorical regarding what is effectively observed and demonstrated. The results support the hypothesis but some questions remain and interpretation should remain more careful in my point of view (please see specific comments).

We thank the reviewer for her/his supportive assessment.

Detailed comments and questions

Introduction

p.2 line 3: Do you have an idea of the relative concentration levels of DOC and DIC in

the study stream?

We have some measurements of $HCO_3^-$ which is the biggest component of DIC in the stream. $HCO_3^-$ values in the stream are generally around 0.1 meq/l.

p.5 line 5 indeed photodegradation has been shown significant on highly brown DOM coming from peatlands (references cited by the authors). I am not sure that it has been reported as important on forested-derived DOM

Indeed, the references cited in the manuscript are from peatlands. We will clarify that photodegradation has been shown significant on DOM coming from peatlands and that we have a forest catchment without peatlands.

Methods

p.4 lines 5-6 This point may be an output from previous research conducted on this well-studied catchment but how the significance of riparian zone contribution has been demonstrated? And quantifications if available would be useful

The flow from the riparian zone to the stream is continuously monitored by thermal cameras and the contribution from different sections of the riparian zone to discharge is measured by dense discharge measurements (salt dilution method) along the stream. This is still work in progress as part of two PhD projects. We know that the hydraulic gradient is towards the stream and that our measurement location is constantly saturated.

p. 4 lines 11-13 provide information about average annual pattern of flow. Similar information about the annual behaviour of DOC concentration and SUVA-254 would be useful to understand the catchment: from Fig. 2 it seems that mean DOC and mean SUVA-254 are maximal in summer low-flow period (increase from Feb to Aug 2014 and from Feb to June 2015). If riparian subsurface flow is the main source of aromatic DOC, I expected this contribution being higher in high flow periods and lower in low flow periods when catchment saturation decreases and therefore minimal DOC and SUVA values in this low flow period

This data is clearly interesting, yet we want to focus, within this manuscript, on diel fluctuations in this manuscript. The reviewer's observations from Fig. 2 are right. We addressed this in another manuscript that is currently under review and that focuses on the event and annual time scale

(Schwab et al., 2017). After reaching a storage threshold during wet conditions (and therefore during high flows) an additional runoff process is playing an important role: subsurface flow / shallow groundwater flow with low DOC and SUVA-254 values.

p. 4 lines 16-17 I think there is an error: unless I am mistaken a variation of 5_C leads to a viscosity change of 2% only. Using Eq. 3 from Schwab et al. (2016):

$\_(T\_)=e\^{3.7188+578.919/(137.546+T\_)}$

If T_ is the temperature in Kelvin degree (as said "T in K"). I think that the 12 to 15% of change of viscosity have been deduced by applying the formula with Temp in Celsius degree, right?

As brief example, for T=15 Celsius deg (288 Kelvin deg) I found Nu=164 Pa.s

for T=20 Celsius deg (293 Kelvin deg) the formula gives Nu=161 = 164 -2%

Yes, there is a mistake in (Schwab et al., 2016). Two minus signs are missing. The equation should be (with T in Kelvin):

$(Nu)=e\^{-3.7188+578.919/(-137.546+T)}$

http://ddbonline.ddbst.de/VogelCalculation/VogelCalculationCGI.exe

With the corrected equation (we performed the calculations based on the correct equation in both papers, but reported the equation not correctly in the WRR paper), a temperature change of 5 °C leads to viscosity changes of 12 % to 15%.

p. 4 lines26-31: Did you compare also the absorbance values at 254 nm from spectro::
lyser and from the lab? (Since Absorbance values are available for the endmembers
- p4 lines 33-34- I suppose that some exist for the stream as well: : :?)

Some grab samples from the stream were analyzed for absorbance 254 nm in the lab. In the figure below we compared the grab samples with the in-situ spectrometer values.

[Figure]

p. 5 lines 1-5 sampling points for the end-members, as well as the stream station and the temperature monitoring point should be located on a map of the site.

We will include a map with the sampling and measurement locations

p. 5 line 3 Regarding the method to sample riparian water, I wonder if the riparian area was effectively fully saturated?

Yes, at the sampling location, the riparian area was fully saturated during the sampling period

On another hand, viscosity of riparian water is calculated from a temperature sensor located in the riparian groundwater at 10 cm depth so I imagine that riparian groundwater remains shallower than those 10 cm depth?

The sampling was done in a saturated area.

Is the water table level in this specific zone monitored (that could help giving the local hydraulic gradient with the stream)?

Unfortunately, the water level was not monitored in this specific zone. Yet, we have TIR images from the area that show how GW enters the stream.

Is riparian water sampled at 6 different depths too or only at 10 cm? Do you observe any vertical variability of DOC and SUVA in this riparian zone as shown for soil water in Fig 5 b,d?

The riparian water was only sampled at 10cm depths.

Regarding Fig 5a, the DOC richness is finally much closer between riparian water and groundwater and low.

Riparian water is likely a mixture between groundwater and soil water components.

p.5 line 13 It is not clear in the following which analyses do use this smoothed SUVA time series (obtained from 3 hours moving window) or the raw time series: p. 5 line 25 "the original time series with the 15 min time intervals" is used to compute the distance to daily average

Indeed, this needs some clarification. The smoothed SUVA time series was used for all the following SUVA analysis and is considered as the original time series. We will clarify this in the revised manuscript.

Results

p.7 The difference in amplitude of diel discharge cycle between dormant and growing period is not characterized but amplitude of the discharge diel cycles seems higher in spring (end of dormant period) than in the growing period? Maybe a scale effect to due difference in base flow?

The diel discharge cycles can be explained by two counteracting processes. The viscosity effect is leading to maxima in the afternoon and evapotranspiration is leading to minima in the afternoon. The interplay between those two processes likely affects the diel amplitude of discharge. The viscosity effect is dominant during the dormant season (discharge maxima in the afternoon) and evapotranspiration is dominant during the growing season (discharge minima in the afternoon).

p. 7 Figure 2: It would help to represent dormant and growing periods on the graph by color or shadings or vertical lines for instance

We will improve this in the revised manuscript.

Figure 5: If possible with the scales, the corresponding values in stream water could be added in (a) and (c) to have in mind the relative position of stream between the end members

We will improve that

Discussion

p. 10 lines 5-9: Correlation between DOC and SUVA_254 fluctuations sounds consistent and I certainly agree with the authors on the value of SUVA_254 or such indices as proxy of DOM composition and properties. However, there is a point absolutely not discussed here: the fact that DOC concentration value is computed by the spectro::lyser algorithm using the absorbance value at 254 (or absorbance at 252 & 255nm). Other absorbance values are obviously included in this concentration estimate but the DOC and SUVA variables used here are somehow both functions of measured absorbance at 254 nm, so that their correlation is not fully surprising: : :At least I feel that it deserves a word in the discussion. See also my comment on (p. 4 lines26-31)

As the second reviewer mentioned, there is a correlation between absorbance at 254nm and DOC, as DOC is calculated (measurement method of the spectrometer) based on absorbance 254nm (AND absorbencies at other wavelengths). SUVA 254 is calculated as the absorbance at 254nm normalized by the DOC concentration (SUVA254 = A254/DOC). Consequently, an increase in SUVA254 is based on an increase in A254 that is larger than the increase in DOC concentration. Therefore on increase in SUVA254 is not (only) based on an increase of DOC in general but on an increase in more aromatic DOC.

We will include this into the discussion to clarify this point.

p. 10 lines 14-15 I feel the rejection of the first hypothesis arrives a little bit fast. The absence of in-stream processes is not fully demonstrated to my opinion. Microbial processes are numerous, here you assume DOC concentration increase due to in-stream production should exhibit a low aromaticity and therefore a low SUVA but i) biological processes that recycle the DOC are numerous enough to lead to complex antagonistic results; ii) keep in mind that SUVA is only a proxy of the complex composition of DOC; iii) and again that is this case SUVA is computed using absorbance properties only .

These are valid points. We will take them into consideration to weaken our statement and include the points i) to iii) into the discussion.

On the other hand, all the conclusions are based on relationships between DOC, SUVA and viscosity which is actually an interpretation of measured temperature variations. Therefore, what is established strictly speaking is that DOC and SUVA variations are correlated with temperature in riparian water isn't it? I wonder if the correlations would have been poorer using for instance stream temperature? And temperature is a factor control of viscosity but also many processes, biological processes,

evapotranspiration:

Indeed, strictly speaking the DOC and SUVA variations are correlated with riparian water temperature. It is also true, that the temperature is controlling other processes. In (Schwab et al., 2016) we already analyzed the difference between viscosity and evapotranspiration. In this manuscript we show that the SUVA maxima are in the afternoon, which is a strong indication for terrestrial DOC input and not for biological processes that could have been affected by stream temperature variations.

If I didn't make a mistake on comment regarding (p. 4 lines 16-17) above, variations of 5_C would induce a change (in viscosity and thus also) in hydraulic conductivity of 2%, which is a very small change, and even if the 10-15% of variations are right, I wonder how significant it is on the flow from this area. If you had an estimate of the range of hydraulic conductivity and of the hydraulic gradient to stream (via measurement of groundwater level) this would help to understand the relative weight of such an increase of the viscosity?

As already explained above, the 10-15% variations are the correct values. Unfortunately, we cannot quantify to hydraulic gradient to the stream. Nevertheless, in our previous paper (Diel discharge cycles explained through viscosity fluctuations in riparian inflow (Schwab et al., 2016, Water Resources Research) we argued, that around 50% of the inflow to the stream are affected by viscosity fluctuations.

Finally there are still missing pieces of discussion:

(p. 8 & p. 9 lines 1-2) Correlation between DOC and SUVA daily variations are stronger during dormant period: why if their diel fluctuations have the same origin (riparian flow)? Would this be related to a change of riparian DOM composition? If so, such change would be visible on the end-members samples? Looking back at Fig 6, it appears to me that this difference is explained actually by stronger in-stream processes that would have during growing season comparable effects to viscosity fluctuations. So that seasonal processes would be also a dominant control, isn't it?

Indeed, we explain that difference by stronger in-stream processes during the growing season. As the viscosity effect / the terrestrial input is still stronger than the instream processes (still a peak in the afternoon), we considered the terrestrial input as the dominant control. The reviewer is right, that during the growing season, the instream processes are also an important (if you want a dominant control) control, but not the most dominant control. We will reconsider our wording. Also concerning on how we handle the first hypothesis.

p. 9 lines 10-11 If riparian water is responsible for diel increase of DOC stream concentration,
I found it surprising that the DOC concentration in the riparian water is finally
rather low compared to soil water in the hillslope

The water in the riparian zone seems to be a mixture of soil water and groundwater. The
groundwater is entering the stream through the riparian zone, as the riverbed consists of relatively
impermeable, solid, unweathered bedrock.

p. 11 lines 1-4 see my suggestion for Figure 5
We will include the reviewer's suggestion.

p. 11 lines 14-20 Schwab et al. (2016) concluded that Q fluctuations during dormant
season was indeed resulting from viscosity changes resulting in variable riparian flow to stream, but
in the growing season, the role of evapotranspiration fluctuations was
dominant (leading to diel cycle inversion). The authors explain the fact that Q and
DOC are not affected by the same processes because of the relative influence of those
processes on the respective "sources" of water and DOC. ET controls Q cycle affecting
the whole catchment storage while viscosity controls the DOC from riparian upper
layers. So at the end, those stream signatures are integrating various catchment processes
and disentangling those processes faces the same issue as distinguishing the
processes that can control seasonal cycles on water quality. Maybe in further studies,
it would be worth to try looking at some other parameters that could play the role of
riparian flow tracers to support further the hypothesis.
We fully agree that this work also open new research avenues outlined by the reviewer.

p. 13 lines 5-14: this answers partially my comment on (p. 4 lines 11-13). However I
found this seasonal pattern quite surprising. In many study, Hillslope subsurface flows
merely active during wet conditions intercept the riparian area flushing somehow their
upper soil layers rich in DOC leading to high DOC concentration (and more aromatic as
well). During low flow, saturated area extension is decreased, and connection between
those DOC sources and the stream can be interrupted. Flow is sustained mainly by
groundwater which is poor in DOC so should lead to minimal DOC concentrations excepted
if autochthonous production increases this DOC concentration. The proposed
interpretation for Weierbach catchment should be discussed regarding general understanding
that have been proposed elsewhere.
It would be interesting to have an idea of the importance of the variations we are looking

at (as percentage of flow/concentration/SUVA mean value). I do not discuss the interest of the topic that has been scarcely studied so far but I think that keeping in mind the relative orders of magnitude of the studied phenomena sounds relevant

We see the reviewer's point and we will better explain the process understanding of the Weierbach catchment. Nevertheless, we do not want to go too much into detail, as this is the topic of a paper that is currently under review and that focuses on the event and season scale (Schwab et al., 2017). This manuscript here, should focus on diel fluctuations.

We will better explain the following aspect: The DOC and SUVA254 values in Figure 2 are daily mean values of days WITH diel fluctuations. This does NOT include days with rainfall-runoff events. During rainfall-runoff events with peaks in discharge, we clearly have DOC and SUVA254 peaks in the stream (coming from fast runoff components and having nothing to do with the viscosity effect), no matter if we are in the growing or the dormant season. The higher discharge during the dormant season shown in Figure 2 in combination with lower DOC and SUVA254 values can be explained by the fact, that during the dormant season, the wetness threshold is reached and the (shallow) groundwater (low DOC, low SUVA254) is connected to the stream.

Conclusion

p. 13 lines 26-29: I wonder if other tracers unrelated to carbon dynamics could be interesting for tracking independently the riparian flow for instance. I would also suggest the use of O2 probes to try catching indirect information on metabolic activity of the stream?

$O_2$ probes would have been very helpful for studying metabolic activity. Unfortunately, no $O_2$ probes were installed in the riparian area.

**Citations**

Schwab, M., Klaus, J., Pfister, L. and Weiler, M.: Diel discharge cycles explained through viscosity fluctuations in riparian inflow, Water Resour. Res., 52(11), 8744–8755, doi:10.1002/2016WR018626, 2016.

Schwab, M. P., Klaus, J., Pfister, L. and Weiler, M.: How runoff components affect the export of DOC and nitrate: a long-term and high-frequency analysis, Hydrol. Earth Syst. Sci. Discuss., 1–21, doi:10.5194/hess-2017-416, 2017.